# Shotgun Proteomics of Co-Cultured Leukemic and Bone Marrow Stromal Cells from Different Species as a Preliminary Approach to Detect Intercellular Protein Transfer

**DOI:** 10.3390/proteomes11020015

**Published:** 2023-04-05

**Authors:** Abraham Josué Nevárez-Ramírez, Ana Laura Guzmán-Ortiz, Pedro Cortes-Reynosa, Eduardo Perez-Salazar, Gustavo Alberto Jaimes-Ortega, Ricardo Valle-Rios, Álvaro Marín-Hernández, José S. Rodríguez-Zavala, Eliel Ruiz-May, José Luis Castrejón-Flores, Héctor Quezada

**Affiliations:** 1Laboratorio de Investigación en Inmunología y Proteómica, Hospital Infantil de México Federico Gómez, Dr. Márquez 162, Doctores, Mexico City 06720, Mexico; 2Unidad Profesional Interdisciplinaria de Biotecnología, Instituto Politécnico Nacional, Av. Acueducto s/n, Barrio La Laguna, Mexico City 07340, Mexico; 3Departamento de Biología Celular, CINVESTAV-IPN, Av Instituto Politécnico Nacional 2508, San Pedro Zacatenco, Mexico City 07360, Mexico; 4División de Investigación, Facultad de Medicina, Universidad Nacional Autónoma de México (UNAM), Circuito interior, Av. Universidad 3000, Ciudad Universitaria, Coyoacán, Mexico City 04510, Mexico; 5Departamento de Bioquímica, Instituto Nacional de Cardiología Ignacio Chávez, Juan Badiano 1, Belisario Domínguez—Sección XVI, Mexico City 14080, Mexico; 6Red de Estudios Moleculares Avanzados, Clúster Científico y Tecnológico BioMimic®, Instituto de Ecología A.C. (INECOL), Carretera Antigua a Coatepec 351, El Haya, Xalapa 91073, Mexico

**Keywords:** leukemia, tumor microenvironment, TSPO, spheroids, proteomics

## Abstract

Cellular interactions within the bone marrow microenvironment modulate the properties of subsets of leukemic cells leading to the development of drug-resistant phenotypes. The intercellular transfer of proteins and organelles contributes to this process but the set of transferred proteins and their effects in the receiving cells remain unclear. This study aimed to detect the intercellular protein transfer from mouse bone marrow stromal cells (OP9 cell line) to human T-lymphoblasts (CCRF-CEM cell line) using nanoLC-MS/MS-based shotgun proteomics in a 3D co-culture system. After 24 h of co-culture, 1513 and 67 proteins from human and mouse origin, respectively, were identified in CCRF-CEM cells. The presence of mouse proteins in the human cell line, detected by analyzing the differences in amino acid sequences of orthologous peptides, was interpreted as the result of intercellular transfer. The transferred proteins might have contributed to the observed resistance to vincristine, methotrexate, and hydrogen peroxide in the co-cultured leukemic cells. Our results suggest that shotgun proteomic analyses of co-cultured cells from different species could be a simple option to get a preliminary survey of the proteins exchanged among interacting cells.

## 1. Introduction

Leukemia is one of the main causes of childhood cancer mortality [1]. The most frequent types are acute lymphoblastic leukemia (ALL) and acute myeloid leukemia (AML) [2], in which two of the biggest challenges are relapse and treatment toxicity [3,4]. It has been proposed that relapse may be related to the survival of subsets of leukemic cells in the bone marrow (BM) microenvironment after initial remission, and that these cells create specialized niches where they resist chemotherapy and acquire leukemia-initiating capacities [5,6]. This altered microenvironment involves the interaction of many types of cells, but the communication between leukemic-initiating cells and mesenchymal stromal cells (MSC) has been studied in more detail [5,6,7].

Some mechanisms of intercellular communication involve the transfer of proteins, vesicles, and organelles, which contributes to drug and stress resistance of leukemic cells [8,9,10,11]. For example, the multifunctional protein galectin-3 of stromal origin induces the production of endogenous galectin-3 in ALL cells, and the intracellular content of this protein positively correlates with tolerance to vincristine and nilotinib [12,13,14]. Moreover, the transfer of membrane-enclosed vesicles from MSC to chronic myeloid leukemia cells protects the latter against imatinib [9], and the mitochondrial transfer from MSC to ALL or AML cells, protects against cytarabine [11,15]. Disruption of the cellular interactions between leukemic cells and the BM is considered a promising therapeutic option [7,16,17,18]. One of the experimental models to study these interactions is the culture of leukemic cells growing around stromal spheroids (3D co-culture), which mimics the hypoxic conditions, the presence of different niches, and the transition between leukemic cells with stem cell features and proliferating blasts [19,20,21].

Mass-spectrometry-based shotgun proteomics allows the identification of individual proteins in complex mixtures by measuring the mass-to-charge ratio (*m*/*z*) of peptide ions to partially reconstruct their amino acid sequence [22]. The preferred strategy to study the protein transfer phenomenon is the Trans-SILAC method in which donor cells are cultured in the presence of heavy lysine and arginine isotopologues until a complete label of the proteome, then donor and recipient cells are co-cultured, and finally labeled proteins are detected in enriched recipient cells [23,24]. However, this method requires expensive culture media and is difficult to use *in vivo*. Therefore, here we used traditional shotgun proteomics focusing on the differences in amino acid residues between homologous mouse and human peptides to explore the possibility of detecting protein transfer in 3D co-cultures.

## 2. Materials and Methods

### 2.1. Cell Culture

The CCRF-CEM human ALL cell line (ATCC CCL-119, Manassas, VA, USA) was grown in RPMI-1640 medium (GIBCO 11875-093, Grand Island, NY, USA), and the OP9 stromal cell line from mouse bone marrow (ATCC CRL-2749, Manassas, VA, USA) in α-MEM medium (GIBCO 41061-029, Grand Island, NY, USA); both supplemented with 10% fetal bovine serum (Corning 35-010 CV, Glendale, AZ, USA) and 1% penicillin–streptomycin (GIBCO 15240-062, Grand Island, NY, USA) in culture flasks at 37 °C, 5% CO_2_, and a humidified atmosphere. Before the cells reached 80% confluency, the CCRF-CEM cells were washed 2× with PBS and diluted; the OP9 cells were trypsinized, washed, and reseeded for their expansion or use in subsequent experiments.

The 3D co-cultures were made as previously reported [19,21]. Briefly, spheroids of OP9 cells were formed in 96-well plates with the bottom of the wells covered with irradiated 1% agarose; to each well, 5 × 10^4^ OP9 cells were added in 100 µL of supplemented RPMI1640 and incubated for 24 h. The resultant spheroids were stable, did not breakdown, or disaggregate if the culture media was exchanged or pipetted up and down. Then, 5 × 10^4^ CCRF-CEM cells in 50 µL of fresh medium were added and incubated for 24 h. After this, the co-cultured CCRF-CEM cells were carefully resuspended (pipetting up and down 8-10 times using a 1 mL tip), the spheroid was transferred to a new tube carrying the minimum necessary volume of media, and the CCRF-CEM cells remaining in the wells were pooled and centrifuged (5 min, 800 rpm, 4 °C); the resultant cell pellet was washed 2× (300 µL of cold PBS each) and used for subsequent experiments or kept at −80 °C until lysis.

The effect of chemotherapeutic drugs on cell viability was addressed by culturing the CCRF-CEM cells as described above and then transferring 2 × 10^4^ cells/well in 200 µL of supplemented RPMI1640 containing 20 nM vincristine (Sutivin, Zurich pharma, Mexico City, Mexico) or 5 µM methotrexate (Traxacord, Accord Farma, Mexico City, Mexico) to 96-well plates. After 24 h, in the presence of vincristine, or 72 h in the presence of methotrexate, cell viability was estimated with the XTT assay (Cell proliferation kit II, Roche, Basel, Switzerland). The effect of hydrogen peroxide was addressed using 1 × 10^5^ cells exposed to 14.6 mM H_2_O_2_ for 1.5 h and cell viability was estimated with trypan blue. Cells without drugs were used as controls.

### 2.2. Flow Cytometry

Co-cultured CCRF-CEM cells (2 × 10^5^) were stained with FITC anti-mouse Sca-1 antibody (Biolegend, San Diego, CA, USA, 122506). Briefly, cells were placed in 200 μL of PBS 1× with 2% FBS on ice and anti Sca-1 was added, mixed, and incubated for 30 min. Then, cells were washed with PBS 1X and centrifuged at 1500 rpm for 5 min. Finally, samples were acquired in a CytoFLEX LX cytometer (Beckman Coulter, Brea, CA, USA). Analysis was performed using CytExpert 2.0 software (Beckman Coulter).

### 2.3. Western Blotting

Cells were lysed with RIPA buffer, and 25 μg of protein per lane were separated by 15% SDS-PAGE. Transfer, hybridization, and visualization were made as previously reported [25]. The primary antibodies were goat polyclonal anti-TSPO (1:1000, abcam, Cambridge, UK, ab92291); rabbit monoclonal anti-galectin 3 (1:1000, abcam, Cambridge, UK, ab76245), and mouse monoclonal anti-actin (1:1000, R&D systems, Minneapolis, MN, USA, MAB8929). The secondary antibodies were anti-goat (Zymed, Waltham, MA, USA, 61-1620), anti-rabbit (Invitrogen, Waltham, MA, USA, G21234), and anti-mouse (Jackson ImmunoResearch, West Grove, PA, USA, 115-035-003).

### 2.4. Enzymatic Activity

Cell pellets were resuspended (5 mg of protein/mL) in 25 mM Tris/HCl pH 7.6, 1 mM EDTA, 5 mM dithiothreitol, and 1 mM phenylmethanesulfonyl fluoride. The cells were disrupted by three cycles of freezing in liquid N_2_ and thawing at 37 °C. Enzyme activities in the lysates were determined at 37 °C in 120 mM KCl, 50 mM Mops, 0.5 mM EGTA, pH 7.2. Total activity of aldehyde dehydrogenase (ALDH) was determined using 50 μg of cellular protein extract and measuring the production of NADH during aldehyde oxidation as previously reported [26]. The activities of lactate dehydrogenase (LDH) and pyruvate kinase (PYK) were determined as previously described using 12.5 μg of cellular protein extract [27].

### 2.5. Proteomic Analysis

#### 2.5.1. Protein Digestion and Fractionation

Cell pellets equivalent to 6 × 10^6^ CCRF-CEM cells, were resuspended in 80 µL of trifluoroacetic acid and incubated at room temperature for 10 min [28]. Lysates were neutralized with 10 volumes of 2M Tris pH 8, and proteins were reduced (10 mM Tris (2-carboxyethyl) phosphine, 5 min, 95 °C) and alkylated (20 mM iodoacetamide, 30 min, room temperature in the dark). Protein concentration was determined by turbidity measurement at 360 nm [28] and then adjusted to 1 μg/μL with 0.2 M Tris pH 8. A total of 1 mg of cellular protein was digested with 10 μg of Trypsin/Lys-C (1:100), Mass Spectrometry Grade (Promega, Madison, WI, USA) for 20 h at 37 °C and 300 rpm. Peptides were desalted (Waters Sep-Pack^®^ cartridges, Milford, MA, USA) and vacuum dried. For high-pH fractionation, dried peptides were reconstituted in 100 μL of 15 mM ammonium hydroxide pH 10, 2% acetonitrile, and bound to 2 mg of C18 resin (ZipTip C18, Merck Millipore, Burlington, MA, USA) [29]. Peptides were separated in nine fractions, each eluted with 100 μL of 2, 5, 7.5, 12.5, 15, 17.5, 20, 35, or 50% ACN in 15 mM ammonium hydroxide, pH 10. The fractions were vacuum dried.

#### 2.5.2. Liquid Chromatography and Mass Spectrometry Analysis

Fractions were dissolved in 20 μL 0.1% formic acid, 5% ACN, and 1 μL was injected into a nanoHPLC system Dionex UltiMate 3000 using a pre-column/peptide trap Acclaim PepMap C18 300 µm × 1.5 cm (Thermo Scientific, Waltham, MA, USA) and a separation column Acclaim PepMap™ 100 75 μm × 15 cm, nano Viper C18, 3 μm, 100A (Thermo Fisher Scientific, Waltham, MA, USA) kept at 40 °C coupled to a quadrupole time-of-flight mass spectrometer (Impact II, Bruker, Billerica, MA, USA). Chromatographic runs were performed at a 250 nL/min constant flow of a mixture of 0.1% (*v*/*v*) formic acid in Milli-Q water (Buffer A), and 0.1% (*v*/*v*) formic acid in acetonitrile HPLC grade (Merck, Darmstadt, Germany) (Buffer B) in a linear gradient of 120 min from 2–40% B, followed by a 10 min ramp from 40–95% B, buffer B was kept at 95% for 10 min, and then returned in 1 min to 2% and kept there for 15 min for column re-equilibration.

Electrospray ionization of the eluted peptides was performed with a CaptiveSpray nanoflow source (Bruker) in positive mode, with a spray voltage of 1500 V, temperature 150 °C, and was assisted by a flow of nitrogen boiled on acetonitrile (0.2 bar). The mass spectra were acquired in a data-dependent acquisition mode and the calibration was performed with the ESI-TOF Tuning mix (Agilent, Santa Clara, CA, USA). For Full MS scan, a range of *m*/*z* 50–2200 was used with a resolution of 60,000 at *m*/*z* 1222. The ion precursors for fragmentation were selected for a fixed cycle time of 3 seg, excluding unassigned and 1+ charge state and excluding after two spectra the same precursor within 0.2 min, unless the intensity of the precursor was more than three times higher than in the previous scan. Ions with charge states ranging from 2+ to 5+ were fragmented, while the collision energy was adjusted between 23–65 ev as a function of the *m*/*z* value.

#### 2.5.3. Protein Identification and Relative Quantification

The Max Quant software (version 1.6.6.0) was used to identify and quantify peptides and proteins using the label-free quantification tool [30]. Searches were performed with the *Homo sapiens* and *Mus musculus* reference proteomes simultaneously (taxonomy: 9606 and 10090) downloaded from UniProt (https://www.uniprot.org/ accessed on 20 July 2022). The following parameters were used: MS tolerance 0.07 Da, MS/MS match tolerance 40 ppm, false discovery rate of 1% at protein and peptide level, Trypsin/P-LysC as a digestion enzyme, the number of missing cleavage sites was set to 2, fixed modification of carbamidomethyl (C), the variable modifications were the oxidation of methionine (M) and N-terminal acetylation. The option “match between runs” was disabled, and the LFQ minimum ratio count was set to 2. In the resulting tables, the gene names assigned to human proteins and peptides were written in upper case letters, and those of mouse, in lower case letters; this allowed differentiation of the species to which a particular protein or peptide belonged. Three independent biological replicates were analyzed, and valid identifications were proteins for which a label-free quantification intensity value higher than zero was assigned in at least two of the three replicates, and at least two peptides were detected. Quality control charts and statistics for mass spectrometry data are shown in Appendix A.

Valid identifications of mouse peptides in human CCRF-CEM cells were based on: (*i*) absence in the human proteome and presence in the mouse proteome according to the Protein BLAST tool (https://blast.ncbi.nlm.nih.gov/Blast.cgi accessed on 20 July 2022) where mismatches should not be I instead of L, and (*ii*) the actual detection of the difference in the mass-to-charge signals between consecutive b- or y-type ions in the CID fragmentation spectrum for at least one of the amino acid residues representing mismatches when the sequences of the corresponding mouse and human peptides with the highest sequence identity were aligned (Appendix A). Valid identifications of mouse proteins in human CCRF-CEM cells were based on the detection of at least two peptides fulfilling criteria (*i*) and (*ii*) in at least two of the three biological replicates. Proteomic datasets have been submitted to the MassIVE repository (MSV000090911 and MSV000090943, https://massive.ucsd.edu/ProteoSAFe/static/massive.jsp, accessed on 15 December 2022).

The DAVID Bioinformatics Resources [31] (https://david.ncifcrf.gov accessed on 20 July 2022) were used for the analysis of functional categories.

### 2.6. Statistical Analysis

For comparisons of cell viability and enzymatic activity, the two-tailed Student’s *t*-test and Microsoft Excel were used. To search for significant differences in protein abundance between CCRF-CEM_monoculture_ and CCRF-CEM_co-culture_, the log2-transformed label-free quantification intensity value and the permutation-based FDR Student’s *t*-test, FDR = 1%, and fold change >1.5 were used, comparing only the proteins identified in the three replicates of both groups with the Perseus software (version 1.6.15.0) [32].

## 3. Results

### 3.1. Leukemic Cells Acquired Significant Resistance to Vincristine, Methotrexate, and Hydrogen Peroxide when Co-Cultured with BM Stromal Cells

After 24 h of 3D co-cultures, the leukemic CCRF-CEM cells migrated towards the OP9 spheroid, and virtually no cells in suspension were observed under the microscope. After agitation and separation of the spheroid, it was observed that the suspended cells had a lymphoid phenotype negative to Sca-1, a stromal marker of mouse cells (Figure 1). Their viability, when exposed to 20 nM vincristine for 24 h, 5 μM methotrexate for 72 h, or 14.6 mM H_2_O_2_ for 1.5 h, was higher than that of CCRF-CEM cells grown in monoculture (Figure 2).

### 3.2. Shotgun Proteomic Analyses of Human Leukemic Cells Co-Cultured with Mouse Stromal Cells May Reveal the Intercellular Transfer of Proteins

We hypothesized that the mouse and human proteomes would be different enough to allow for the detection of mouse peptides, from the OP9 cells, present in co-cultured human CCRF-CEM cells, revealing the intercellular protein transfer phenomenon. The stringent criteria described in the Materials and Methods section led to the identification of 67 mouse proteins in co-cultured CCRF-CEM cells (Appendix A) suggesting that these proteins may have been transferred from MSC to leukemic blast cells during the co-culture period. As expected, no mouse proteins were detected in CCRF-CEM cells grown in monoculture. Regarding the human proteins, 1513 and 2086 were identified in CCRF-CEM cells grown in co-culture and monoculture, respectively.

Functional enrichment analysis of the 67 mouse proteins presumably transferred to human cells indicated that the most accurately represented biological processes were related to cellular responses to interleukin-7, drugs, hydrogen peroxide, or endoplasmic reticulum stress, and other processes such as glycolysis, cell shape and motility, or actin dynamics (Figure 3A). This agrees with the representation of molecular functions such as oxidoreductase and protein disulfide isomerase enzymatic activities, binding to monosaccharides and ADP, or binding to phospholipids, Ca^2+^ and actin (Figure 3B). Regarding the cellular components, the proteins were mainly located in the mitochondria, cytosol, endoplasmic reticulum, and cell surface (Figure 3C).

Validation of these proteomic results by Western blot may be inaccurate because antibodies can recognize both the mouse and the human antigens. Still, the TSPO protein was among the valid identifications of mouse proteins in co-cultured CCRF-CEM cells (Appendix A) and Western blot analysis indicated that its abundance was higher in cells harvested from co-cultures than from monocultures (Figure 4A). The used anti-TSPO antibody reacts with the mouse protein, but not with the human protein according to the vendor, which agrees with the lack of signal in extracts from human BM stromal cells (HS5 cell line), astrocytes (SVGp12 cell line), or leukemic blasts (CCRF-CEM cells, in Figure 4B), suggesting that the protein detected in co-cultured CCRF-CEM cells may be of murine origin. The galectin-3 protein was also among the presumably transferred proteins (Appendix A), which agrees with the previously reported transfer of this protein from mouse OP9 to human leukemic cells during co-cultures [12] and with the increased abundance observed in co-cultured cells (Figure 4A); however, the anti-galectin-3 antibody could not differentiate between the murine and human proteins because the signal was detected in human HS5 and SVGp12 cells (Figure 4B).

The mouse mitochondrial aldehyde dehydrogenase ALDH2 was also detected in co-cultured CCRF-CEM cells (Appendix A), which agrees with an increase in the total ALDH activity in co-cultured cells compared to cells grown in monoculture (Figure 5A). The proteomic analysis also identified the mouse glycolytic enzymes lactate dehydrogenase (LDH) and pyruvate kinase (PYK) in the co-cultured human CCRF-CEM cells (Appendix A). For these two proteins, the average activity was higher in co-cultured cells, but the high variability hampered the detection of significant differences (Figure 5B,C).

Unexpectedly, no significant differences were found in the abundance of the 1409 human proteins that were detected in both the co-cultures and monocultures of the CCRF-CEM cells. This may be due to the high variability observed in the label-free quantification intensity values of proteins detected in co-cultured cells.

## 4. Discussion

The reduced sensitivity to chemotherapeutic drugs and oxidative stress of leukemic cells co-cultured with stromal cells has been reported previously [11,12,13,14,15,33,34,35]. Co-cultures of mouse and human cells have been used to study this effect [11,14,34,35], and the intercellular transfer of mitochondria, vesicles, and galectin-3 from MSC to leukemic cells, has been associated with this protection in experimental settings where transfer goes from human to human [9,15,16,36] and from mouse to human cells [11,14,15,16]. In these reports, cells interacted in 2D co-cultures, in which leukemic cells grew on top of a monolayer of stromal cells; only the work of Kolba and coworkers, using the Trans-SILAC method and chronic myeloid leukemia cells, provided a proteome-scale map of the transferred proteins [9]. Therefore, herein, we reported the use of shotgun proteomics to get a preliminary picture of the proteins that may have been transferred from MSC to leukemic cells in a 3D co-culture system that resembles the bone marrow microenvironment of ALL.

Our experimental approach is simple but highly dependent on the absence of contaminant OP9 stromal cells detached from the spheroids in the preparation of co-cultured CCRF-CEM cells. The absence of the stromal marker Sca-1 in co-cultured leukemic cells (Figure 1) indicated that the level of contamination was indeed very low (0.7 ± 0.5% of positive reads without clustering on a specific region of SSC values). Moreover, the mass spectrometer was operated in the data-dependent acquisition mode, which is biased to pick peptides with the strongest signal, most likely coming from the predominant human cells. For these reasons, we consider that the probability that the origin of the 67 mouse proteins (representing 4.2% of valid identifications in co-cultured CCRF-CEM cells) was the 0.7% contaminant mouse OP9 cells, was very low.

Other sources of contamination could be cell remnants. In this regard, it has been reported that the dissociation of cells from the BM, spleen, or lymph nodes caused fragmentation of resident macrophages, and that these fragments, attached to hematopoietic cells, contaminated single-cell suspensions [37]. In that work, the dissociation was made using 70 μm strainers, a mechanical dissociator, or by flushing the BM through a 27G needle prior to filtration (40-micrometers) [37]. In the present work, the co-cultured leukemic cells were resuspended by carefully pipetting up and down eight to ten times using a 1 mL tip; the spheroid remained intact when observed under the microscope, and as indicated by the absence of Sca-1, resuspension of leukemic cells did not result in evident dissociation or fragmentation of the adherent OP9 cells. When the co-cultured spheroids were trypsinized and mechanically disrupted, the cell viability was 91 ± 4%. This result suggested that the mild mechanical dissociation of leukemic cells from the 3D co-culture did not cause significant cell membrane damage and protein leakage from the adherent OP9 cells. Such leakage could be another source of contamination. It was assumed that the two washes of co-cultured CCRF-CEM cells before lysis minimized the possibility of contamination with murine-secreted proteins and extracellular vesicles attached to the surface of human cells.

Contaminant peptides may remain in the HPLC-MS/MS analytical instruments after the injection of samples. By analyzing results from alternate injections of samples from the OP9 and CCRF-CEM cells grown in monoculture, it was determined that contaminant peptides led to less than seven valid identifications of proteins from one species in the samples of the other, none of which were considered for further analysis or included in the Appendix A.

Cells can exchange proteins through structures such as gap junctions, extracellular vesicles, or tunneling nanotubes [8,9,10,38,39]. The terms related to binding to actin, Ca^2+^, and phospholipids in Figure 3B, resulted from the identification of proteins associated to the S100 and annexins complexes, probably reflecting the transfer of vesicles [40]. Other proteins associated with the actin cytoskeleton and regulation of cell shape in Figure 3A, probably reflect transfer through the actin-rich tunneling nanotubes [8,9,10,24].

Among the presumably transferred proteins, TSPO, carbonyl reductase 2, peroxiredoxin-1, glutathione S-transferase Mu 1, or aldehyde dehydrogenase ALDH2 have protective functions against drugs and cellular stress, which correlates with the observed resistance to vincristine, methotrexate, and hydrogen peroxide of co-cultured CCRF-CEM cells. Activation of the interleukin-7 pathway leads to increased survival of leukemic cells [41]; it is likely that certain mouse proteins contributed to activate this pathway in human cells (Figure 3A).

About one fifth of the presumably transferred proteins of Appendix A have preferential mitochondrial localization according to the UniProt database (https://www.uniprot.org/ accessed on 20 July 2022). The representation of GO terms related to mitochondria in Figure 3C agrees with the reported mitochondrial transfer from mouse MSC to human leukemic cells [11,15,16]. Representation of the GO term “Response to drug” (Figure 3A) and proteins related to the redox homeostasis (Appendix A), agree with the results of the proteomic study of Kolba and coworkers, which used chronic myeloid leukemia cells and human MSC [9]. Detection of the galectin-3 protein of mouse origin agrees with the reported transfer of this protein from mouse MSC to human ALL cells [12].

Although several proteins in the Appendix A are related to the glycolytic pathway, our results suggest that the transfer of LDH and PYK did not happen to such an extent that significantly increased the already high amounts of the active forms of these enzymes in co-cultured cells (Figure 5B,C). Regarding the increased activity of ALDH (Figure 5A), besides the transfer of ALDH2, it was also possible that the co-culture conditions induced the synthesis of the endogenous ALDH isoforms.

The mitochondrial translocator protein (TSPO) is abundant in glial cells, such as the SVGp12 cell line used in Figure 4B, and has been involved in the maintenance of the mitochondrial membrane potential, steroid hormone synthesis, opening of the mitochondrial permeability transition pore, and apoptosis [42,43,44]. Lymphoid cells express low levels of TSPO, but when its expression was induced in Jurkat cells, the transfected clones significantly increased cell viability in the presence of H_2_O_2_ [45] and showed lower levels of apoptosis and superoxide anions when irradiated with ultraviolet light [46]. This protein also contributes to resist the effects of reactive oxygen species in mouse retina [47], and its expression has been proposed as a prognostic marker in patients with chronic lymphocytic leukemia, in whom lower levels of TSPO correlated with a better response to treatment [48]. Here, we demonstrate that the abundance of this protein increased in leukemic cells when co-cultured with stromal cells, likely as a result of intercellular protein transfer events.

One of the limitations of the present study is the need of cells from different species which entails the possibility that the observed results do not replicate in the model of two cell types of one origin. In this regard, the co-culture of mouse MSC and human leukemic cells has been used to study the protection against chemotherapeutic drugs and intercellular mitochondrial transfer [11,12,14,15,16,34,35]. These effects have been replicated in co-cultured human cells [9,10,33,36]. Indeed, OP9 cells were superior to human MSC, grown from primary human bone marrows, in promoting proliferation and providing protection against vincristine to human ALL cells [14].

Other limitations of our experimental approach are the inability to address the possibility of transfer of proteins with identical amino acid sequences, and the dependency of species-specific antibodies for validation with immunoassays. Regarding validation, targeted proteomics using the selected reaction monitoring/multiple reaction monitoring (SRM/MRM) methodology is an alternative to avoid cross reaction of antibodies [49].

Proteoforms significantly contribute to proteome complexity [50]. The abundance of the many proteoforms and their interactors may be different in donor and recipient cells. For example, activation of different kinases may lead to distinct phosphorylation patterns of the murine TSPO protein in OP9 and CCRF-CEM cells, which could result in differential regulation of the TSPO activity. Once the transfer of a protein has been detected, targeted analyses of that specific protein should be conducted to compare the abundance of the different proteoforms in donor and recipient cells. To this end, top-down proteomics can be used [50] after enrichment of the transferred protein from co-cultured recipient cells, by affinity purification using species-specific antibodies or tagged proteins expressed in donor cells.

## 5. Conclusions

Our results suggest that an in-depth examination of the sequence and fragmentation pattern of peptides detected in shotgun proteomics analyses of co-cultured cells from different species, may be a simple option to study the intercellular protein transfer phenomenon. This approach could easily be used *in vivo* in leukemia xenograft models. Our preliminary study is a proof of concept, and further research is needed to address the suitability and type of bias of the herein proposed methodology. A direct comparison with the Trans-SILAC method will certainly shed some light on this issue.

## Figures and Tables

**Figure 1 proteomes-11-00015-f001:**
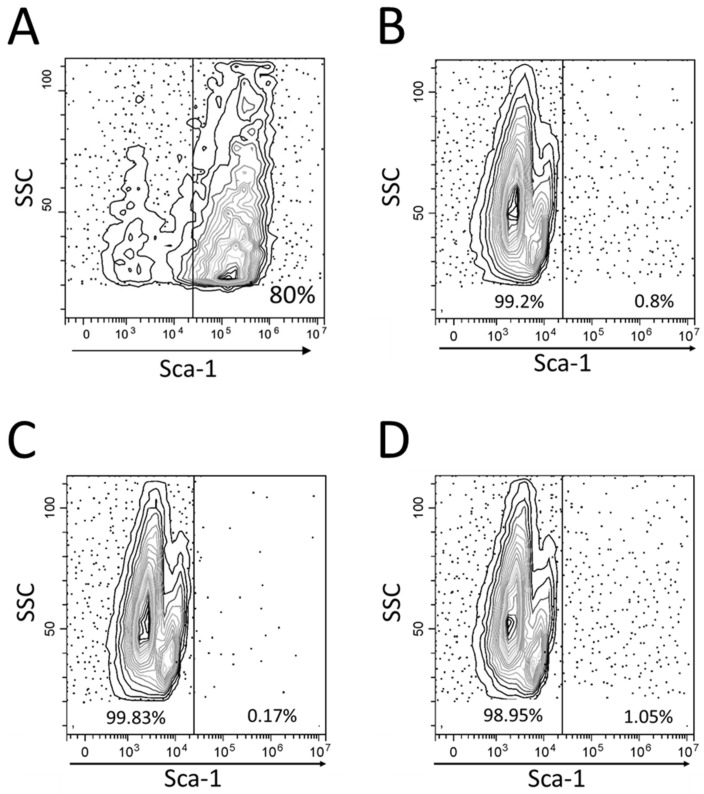
The human CCRF-CEM leukemic cells were recovered with high purity from the 3D co-cultures. (**A**) OP9 cells express the mouse stromal marker Sca-1. (**B**–**D**) The mouse Sca-1 marker is not detected on non-adherent cells harvested from three independent co-cultures.

**Figure 2 proteomes-11-00015-f002:**
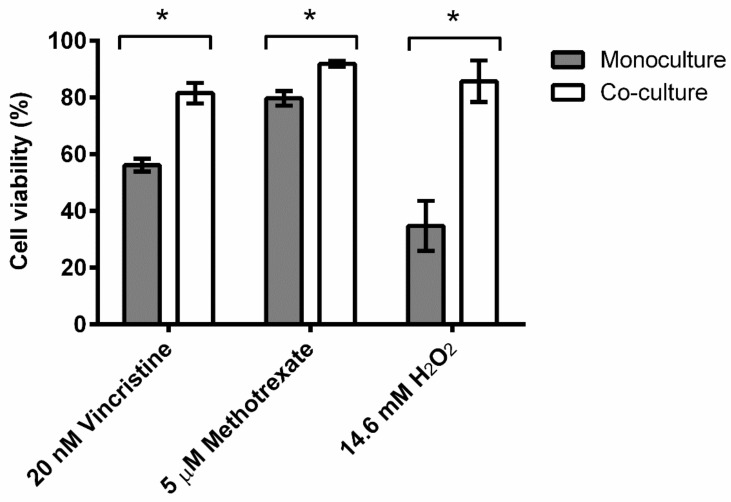
Co-cultured CCRF-CEM cells showed lower sensitivity to the cytotoxic effects of chemotherapeutic drugs and hydrogen peroxide than cells grown in monoculture. Cells were exposed to vincristine for 24 h, methotrexate for 72 h, or H_2_O_2_ for 1.5 h. Results are presented as the mean and standard deviation values of three independent cell cultures. * *p* < 0.01, Student’s *t*-test.

**Figure 3 proteomes-11-00015-f003:**
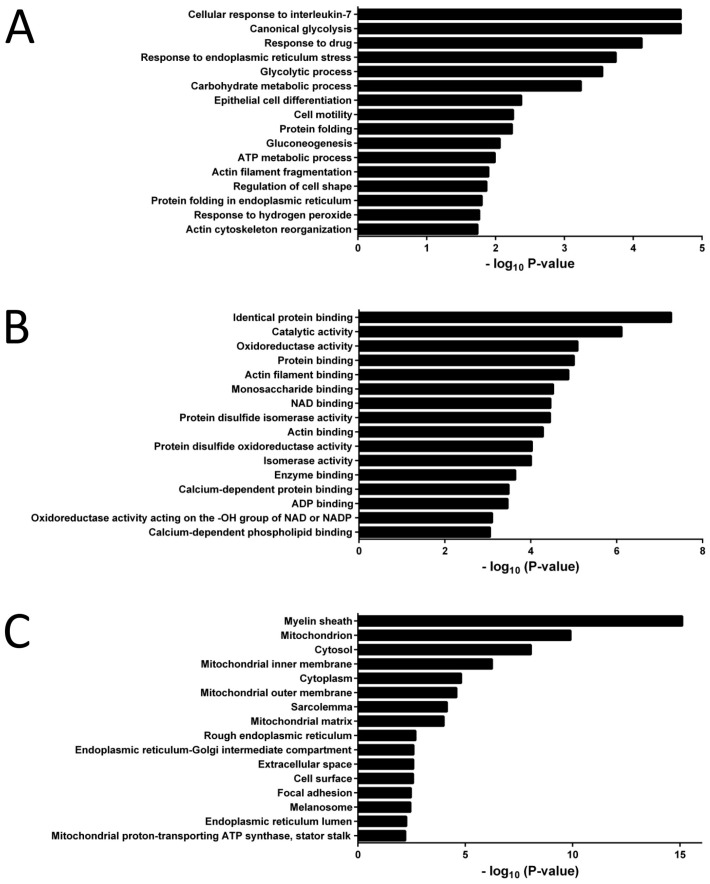
Most accurately represented GO terms among the mouse proteins detected in co-cultured CCRF-CEM cells (Appendix A). (**A**) Biological process. (**B**) Molecular function. (**C**) Cellular component. The DAVID Bioinformatics Resources web site, and the “Functional Annotation Chart” tool were used (https://david.ncifcrf.gov accessed on 27 July 2022).

**Figure 4 proteomes-11-00015-f004:**
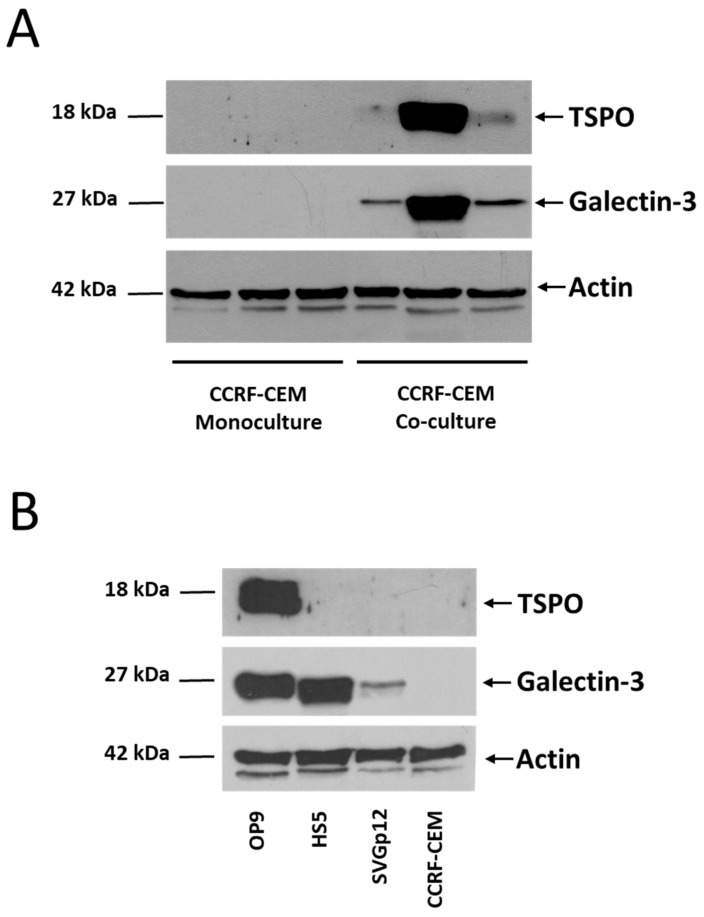
The abundance of the TSPO and galectin-3 proteins is higher in co-cultured CCRF-CEM cells than in cells grown in monoculture. (**A**) Three independent cultures of each condition were compared. (**B**) The primary anti-TSPO antibody recognizes preferentially the mouse protein. OP9, mouse cell line established from bone marrow mesenchymal stromal cells; HS5, human cell line established from bone marrow stroma; SVGp12, human cell line established from astroglia; CCRF-CEM, human cell line established from T-lymphoblasts. The uncropped images of Western blot experiments are shown in the Appendix A.

**Figure 5 proteomes-11-00015-f005:**
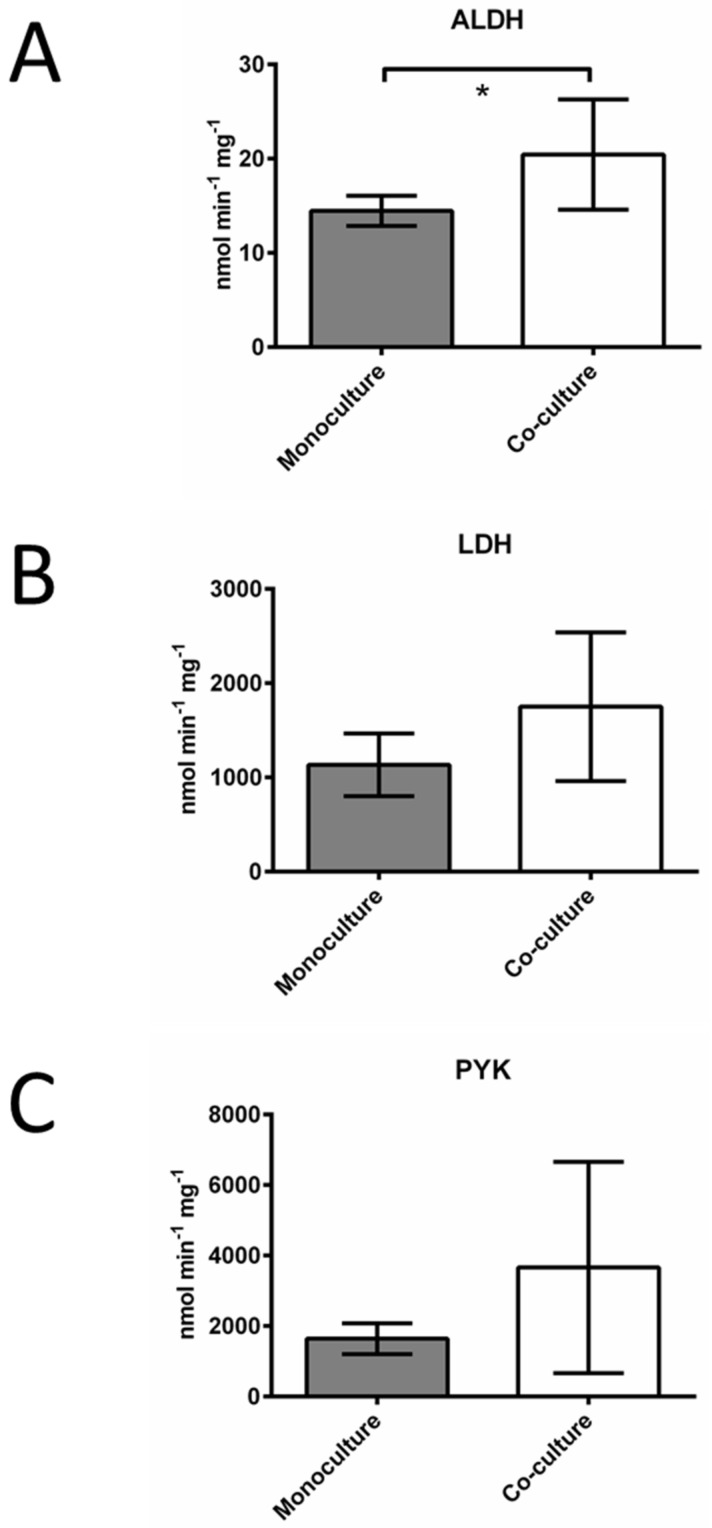
Enzymatic activities of aldehyde dehydrogenase (ALDH, **A**); lactate dehydrogenase (LDH, **B**) and pyruvate kinase (PYK, **C**) determined in cell extracts obtained from CCRF-CEM cells. These enzymes are among the mouse proteins detected in the proteomic analysis of co-cultured CCRF-CEM cells. Mean and standard deviation values of at least three independent cell cultures are shown, * *p* = 0.036.

## Data Availability

Data presented in this study are openly available in the MassIVE repository (https://massive.ucsd.edu/ProteoSAFe/static/massive.jsp, accessed on 15 December 2022) divided in two data sets. For the CCRF-CEM cells, doi:10.25345/C5X63B95P, reference number: MSV000090911, FTP Download Link: ftp://MSV000090911@massive.ucsd.edu (accessed on 15 December 2022), data set identifier: MSV000090911, username: Hector_Quezada_1 and password: Nevaresetal2022. The second part contains data for the OP9 cells, doi:10.25345/C5NK3694V, reference number: MSV000090943], FTP Download Link: ftp://MSV000090943@massive.ucsd.edu (accessed on 20 December 2022), data set identifier: MSV000090943, username: Hector_Quezada_1 and password: Nevaresetal2022b.

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
