# Peer review of "Shotgun Proteomics of Co-Cultured Leukemic and Bone Marrow Stromal Cells from Different Species as a Preliminary Approach to Detect Intercellular Protein Transfer"

_proteomes, 2023, doi:10.3390/proteomes11020015_

Round 1

Reviewer 1 Report

Review Nevárez-Ramírez et al.

In their work, Nevárez-Ramírez and coworkers are attempting to determine the exchange of proteins in a human mouse cell co-culture by determining the exchange based on the species association of the peptides found. The peptides are mapped to proteins and describe 67 proteins that are exchanged between the two different cell types. The data is then supported by western blotting.

Major criticism 

The method described in this manuscript is supposed to be an alternative to trans-SILAC experiments but without the higher experimental requirements of the SILAC technique. To make this point the authors have to show more data about the quality of their peptide data. 

  1. It is necessary to show for at least 10 annotated peptide spectra that the quality of the measurement is sufficient to distinguish the peptides with enough certainty. Provision of the dot product to demonstrate the identity of the peptide spectra.
  2. A plot of covariances at peptide and protein levels. Some QC has to be done, like correlation profile between replicates (pearson or spearman), normalization and protein distribution, PCA. Z-score, heatmaps showing the quantification across replicates and variances on the dysregulated proteins for mouse and human.
  3. The authors state: “The preferred strategy to study the protein transfer phenomenon is the Trans-SILAC method in which donor cells are cultured in the presence of heavy lysine and arginine isotopologues until complete label of the proteome, then donor and recipient cells are co-cultured, and finally labeled proteins are detected in enriched recipient cells [23,24]. However, this method requires expensive culture media and is difficult to use in vivo. Therefore, here we used traditional shotgun proteomics focusing on the differences in amino acid residues between homologous mouse and human peptides to explore the possibility of detecting protein transfer in 3D co-cultures.” To truly show that their method is a valid alternative to trans-SILAC they have to provide a comparison of the two methods. Especially since their methodology creates a bias for peptides that are not common and thus proteins that produce more diverse peptides than the host proteome. 
  4. The peptide quality cut-off for the peptides should be set to 1% not only the protein cut-off.
  5. The biological data provided by the authors in figure 4 A shows very low reproducibility between the experiments. The signal of Galectin-3 ranges from hardly a band to a overexposed band. The same is true for TSPO. 
  6. The authors should provide some additional evidence that the proteins are indeed exchanged. Possibilities would be the use of GFP fusion proteins that can be followed during the exchange. 
  7. The connection to the metabolic enzymes seems very speculative and the direct connection to the proteomics data is not given. If indeed the mitochondrial functions are modulated then the authors should show that the number of mitochondria stays the same or are changed in response to the exchange. 
  8. The discussion is too long for the evidence provided in the manuscript and should be shorted.

Minor criticism

1. Line 265 anti-galcetin-3 antibody 

typo (should be galectin)

Author Response

REVIEWER 1

Major criticism 

The method described in this manuscript is supposed to be an alternative to trans-SILAC experiments but without the higher experimental requirements of the SILAC technique. To make this point the authors have to show more data about the quality of their peptide data.  

  1. It is necessary to show for at least 10 annotated peptide spectra that the quality of the measurement is sufficient to distinguish the peptides with enough certainty. Provision of the dot product to demonstrate the identity of the peptide spectra.

Many thanks for this pertinent observation. To show that peptide identification was made with enough certainty, we have selected 23 mouse-exclusive peptides detected in the suspension of co-cultured CCRF-CEM human cells, and their MS/MS fragmentation spectra are shown in the new Supplementary Figure S1. Additionally, we added a new column to the Supplementary Table S1 indicating the “score” value assigned by the MaxQuant software. 

  1. A plot of covariances at peptide and protein levels.

Thanks, this plot has been included in the new Supplementary Figure S1.

  1.  Some QC has to be done, like correlation profile between replicates (pearson or spearman), normalization and protein distribution, PCA. Z-score, heatmaps showing the quantification across replicates and variances on the dysregulated proteins for mouse and human.

Thanks for this suggestion. The quality control charts and statistics for the identification of proteins and peptides are shown in the new Supplementary Figure S1. This Table is mentioned in the text in lines 186-187 of the version with tracked changes.

  1. The authors state: “The preferred strategy to study the protein transfer phenomenon is the Trans-SILAC method in which donor cells are cultured in the presence of heavy lysine and arginine isotopologues until complete label of the proteome, then donor and recipient cells are co-cultured, and finally labeled proteins are detected in enriched recipient cells [23,24]. However, this method requires expensive culture media and is difficult to use in vivo. Therefore, here we used traditional shotgun proteomics focusing on the differences in amino acid residues between homologous mouse and human peptides to explore the possibility of detecting protein transfer in 3D co-cultures.” To truly show that their method is a valid alternative to trans-SILAC they have to provide a comparison of the two methods. Especially since their methodology creates a bias for peptides that are not common and thus proteins that produce more diverse peptides than the host proteome. 

This is an important observation, and the reviewer is correct that a direct comparison between our method and Trans-SILAC is necessary to assess the suitability and type of bias of our approach. However, our study represents an initial step towards exploring the possibility of detecting the intercellular protein transfer using a simple shotgun proteomics experiment. We recognize that more work is needed to establish the validity of our methodology as an alternative to Trans-SILAC, as discussed in lines 418-421 of the revised version with tracked changes.

  1. The peptide quality cut-off for the peptides should be set to 1% not only the protein cut-off.

Many thanks for this pertinent observation. The peptide cut-off was indeed set to 1%. This is mentioned in line 176 of the version with tracked changes and is also specified in the mqpar.txt file submitted to the MassIVE repository doi:10.25345/C5X63B95P, reference number: MSV000090911. 

  1. The biological data provided by the authors in figure 4 A shows very low reproducibility between the experiments. The signal of Galectin-3 ranges from hardly a band to a overexposed band. The same is true for TSPO. 

Although 3D culture systems are attractive due to their ability to better reproduce cell-to-cell interactions, its use for methodologies that require a high number of cells, involves the need to process several batches of laborious handling. We believe that this contributed to the variable results. Despite implementing several steps to process the batches in the same way, the metabolic state of the 3D co-cultured CCRF-CEM was consistently heterogeneous, resulting in variability in Western blot experiments, enzymatic activity determinations (Figure 5), and the separation of corresponding dots in the PCA (new Supplementary Figure S1). This effect was not observed in monocultures. Nevertheless, the detection of both proteins, TSPO and galectin-3, in three independent biological replicates can be interpreted in the context of intercellular transfer or proteins as a yes/no experiment. From this perspective, the results in Figure 4A are in line with the data from our proteomics analysis. While we recognize the inherent variability in in vitro systems, we are confident that this variability did not detract from the main objective of our study.

  1. The authors should provide some additional evidence that the proteins are indeed exchanged. Possibilities would be the use of GFP fusion proteins that can be followed during the exchange. 

Thank you for the suggestion. We agree that the experiment using fluorescently tagged proteins will indeed demonstrate the exchange of proteins. It is worth noting that previous studies have reported protein transfer between mouse and human cells, including galectin-3 from mouse cells to human acute lymphoblastic leukemia cells [reference 14] and mitochondrial transfer from mouse cells to human acute myeloid leukemia cells in 2D co-cultures [references 11,15]. These findings are consistent with our proteomics data and support the possibility of protein transfer in our 3D co-culture system. Our identification of mouse proteins in human cells with mouse-exclusive peptides (supplementary Table S1) and detection of the TSPO protein from mouse in human cells (Figure 4A) also support this assumption.

We will express fluorescently labeled proteins in the OP9 cells and track them in CCRF-CEM cells after the 3D co-culture. Unfortunately, these techniques are laborious and will not be possible to implement expeditiously. As stated before, the aim of this investigation is to provide a proof of concept that it may be possible to detect the intercellular transfer of proteins with a simple shotgun proteomics experiment with some caveats. For this reason, along the revised manuscript, we mention the exchange of proteins as a possibility, recognizing that this is a preliminary study.  Please see lines 308-309, 387 and 418-421 of the version with tracked changes.

  1. The connection to the metabolic enzymes seems very speculative and the direct connection to the proteomics data is not given. If indeed the mitochondrial functions are modulated then the authors should show that the number of mitochondria stays the same or are changed in response to the exchange. 

We apologize for any lack of clarity regarding this topic that may have made it sound speculative. The lack of correlation between the putative transfer of lactate dehydrogenase (LDH) and pyruvate kinase (PYK) suggested by our proteomic analysis, and the corresponding enzymatic activities (Figure 5 B, C and Supplementary Table S1) is a negative result. However, we believe that it can contribute to the understanding of the metabolic impact of the possible transfer of these enzymes, not increasing the already high activity of these glycolytic enzymes in the recipient cells. This is mentioned in lines 370-373 of the version with tracked changes.

Our proteomic analysis also indicates the putative transfer of the mitochondrial ALDH2 isoform which may correlate with the increased activity of aldehyde dehydrogenase (ALDH) in co-cultured leukemic cells (Figure 5A). This is an original contribution that, to the best of our knowledge, has not been reported previously and which may contribute to the understanding of the metabolic impact of the putative mitochondrial transfer.

For the above reasons, we believe it may be worth sharing our metabolic results with the readers. However, as the reviewer correctly states, more experimental evidence is needed to establish a firm connection between protein transfer and metabolic functions. A possible approach would be to label OP9 cells´ mitochondria using fluorescent dyes and track them in the co-cultured CCRF-CEM. Unfortunately, implementing these techniques in a timely manner is not feasible. At this point, we want to report a preliminary study about the possibility of detecting the intercellular transfer of proteins with a simple shotgun proteomics experiment.

  1. The discussion is too long for the evidence provided in the manuscript and should be shorted.

 Thanks for the observation. The discussion is indeed long but we tried to clearly address the potentialities and limitations of the proposed methodology. We slightly reduced the discussion removing lines 344 and 345 of the version with tracked changes.

 Minor criticism

  1. Line 265 anti-galcetin-3 antibody 

 typo (should be galectin)

Thanks, the typo has been corrected. Line 268 of the version with tracked changes.

Reviewer 2 Report

In this paper, Nevarez-Ramirez et al., explore the possibility of detecting proteins transfer in 3D co-cultures from different species (mouse and human) by using shotgun proteomics. The paper is interesting and their results are valuable. Overall, this approach is original and correct. However, there are important issues that have to be addressed before accepting the paper.

Major revision:

Supplementary figures: Figures need to improve.

1.     Hand notes should be replaced.

2.     Ladder is missing in the gel (they are writing by pen). Gels without ladder should not be accepted. Please, provide Western Blot gels with ladder.

Minor revision

Results:

Validation of the proteins should be described in a flow. They are confusing; I would like to know how many proteins are have significant differences.

References: some of them should be updated. For example, number 2 or number 45,46.

Author Response

REVIEWER 2

In this paper, Nevarez-Ramirez et al., explore the possibility of detecting proteins transfer in 3D co-cultures from different species (mouse and human) by using shotgun proteomics. The paper is interesting and their results are valuable. Overall, this approach is original and correct. However, there are important issues that have to be addressed before accepting the paper.

Major revision:

Supplementary figures: Figures need to improve.

  1. Hand notes should be replaced.

Thanks for this pertinent observation. A new supplementary Figure S2 was made without hand notes and showing images of only one exposure time for each membrane.

  1. Ladder is missing in the gel (they are writing by pen). Gels without ladder should not be accepted. Please, provide Western Blot gels with ladder.

Thank you for your suggestion, and we agree with the reviewer that the presence of the ladder is essential for Western blot analyses. In our Western blot experiments, we used X-ray films for chemiluminescence detection, which unfortunately does not allow for the detection of the ladder, unlike with ChemiDoc imagers where the signal of the ladder is directly detected from the membrane. To address this issue, we placed the membrane on top of the film and marked the ladder positions guided by the small diagonal cut made in a bottom corner. However, in our opinion, the use of X-ray films for chemiluminescence detection remains a valid alternative for Western blot analysis. In the new Supplementary Figure S2, we indicate the position of the ladder based on the marks made on the film. We hope this revised figure is suitable for publication.

Minor revision

Results:

Validation of the proteins should be described in a flow. They are confusing; I would like to know how many proteins are have significant differences.

We apologize for the lack of clarity regarding protein validation. Several changes were made to avoid confusions, including re-arrangement of protein presentation in Figure 4A and changes in the text within lines 256-270 of the version with tracked changes, where the relevant cell lines have been indicated.

Regarding the number of proteins that showed significant differences in abundance, unexpectedly, our proteomic analysis did not reveal the presence of differentially regulated proteins. We believe that this was related to the high variability observed in the results of the co-cultured CCRF-CEM cells. This is mentioned in lines 293-296 of the version with tracked changes.

References: some of them should be updated. For example, number 2 or number 45,46.

Thanks for this suggestion. Reference 2 has been updated. The new reference is:

Reinhardt, D.; Antoniou, E.; Waack, K. Pediatric Acute Myeloid Leukemia-Past, Present, and Future. J. Clin. Med. 2022, 11, 504. doi: 10.3390/jcm11030504.

References 45 and 46 describe results about the protective effect of the TSPO protein in leukemic cells against the cytotoxic effect of hydrogen peroxide and free radical damage induced by UV light. We could not find more recent reports describing these effects. For these reasons, we believe that it is convenient to keep references 45 and 46 for the discussion of the possible physiological relevance of the intercellular exchange of the TSPO protein (lines 376-388 of the version with tracked changes).

Reviewer 3 Report

The manuscript by Nevarez-Ramirez et al titled "Shotgun proteomics of co-cultured leukemic and bone marrow stromal cells from different species as a preliminary approach to detect intercellular protein transfer" utilized a novel 3D co-culturing system of human and mouse cells to study intercellular communication. Subsequent proteomics analysis revealed proteins that potentially transferred between cell types. While the experimental setup in principle could allow for detection of proteins from distinct cell types, it was not clean enough in reality to substantiate the main findings. Therefore I suggest the authors to consider my comments below and revise the current study significantly.

Specific comments:

Figure 1: The whole premise of the current experimental paradigm hinges on the hypothesis that human CCRF-CEM cells can be isolated cleanly after co-culturing with mouse OP9 cells in a 3D culture system. And the flow cytometry analysis in Figure 1 largely supports this notion, with only 0.8% of the isolated human cell population being mouse OP9 cells. 

The authors leveraged this system for proteomics analysis of this isolated population and identified 1513 human proteins and 67 mouse proteins. A conclusion was drawn that these 67 mouse proteins were transferred from mouse OP9 cells to human CCRF cells during co-culture. However, an alternative, more parsimonious explanation for the presence of these 67 mouse proteins is that they came from the 0.8% of the sample that are mouse OP9 cells. The current study did not have convincing control data to rule this possibility out.

I suggest the following experiment to distinguish between the two possibilities: Express a fluorescent protein in either the human or mouse cell type and sort out only the human cells from the sample by FACs.

Figure 2: The authors nicely showed that co-cultured cells are more resistant to insults. Indeed, this validates the 3D culture system as a novel and useful tool to study drug resistance in cancer.

Figure 4: The authors validated two proteins identified from the proteomics analysis, albeit with the caveat laid out above. While the authors nicely include three replicates in the figure, they are quite variable. I believe adopting the revised setup will alleviate this inconsistency.

Author Response

REVIEWER 3

The manuscript by Nevarez-Ramirez et al titled "Shotgun proteomics of co-cultured leukemic and bone marrow stromal cells from different species as a preliminary approach to detect intercellular protein transfer" utilized a novel 3D co-culturing system of human and mouse cells to study intercellular communication. Subsequent proteomics analysis revealed proteins that potentially transferred between cell types. While the experimental setup in principle could allow for detection of proteins from distinct cell types, it was not clean enough in reality to substantiate the main findings. Therefore I suggest the authors to consider my comments below and revise the current study significantly.

 Specific comments:

 Figure 1: The whole premise of the current experimental paradigm hinges on the hypothesis that human CCRF-CEM cells can be isolated cleanly after co-culturing with mouse OP9 cells in a 3D culture system. And the flow cytometry analysis in Figure 1 largely supports this notion, with only 0.8% of the isolated human cell population being mouse OP9 cells. 

The authors leveraged this system for proteomics analysis of this isolated population and identified 1513 human proteins and 67 mouse proteins. A conclusion was drawn that these 67 mouse proteins were transferred from mouse OP9 cells to human CCRF cells during co-culture. However, an alternative, more parsimonious explanation for the presence of these 67 mouse proteins is that they came from the 0.8% of the sample that are mouse OP9 cells. The current study did not have convincing control data to rule this possibility out.

 I suggest the following experiment to distinguish between the two possibilities: Express a fluorescent protein in either the human or mouse cell type and sort out only the human cells from the sample by FACs.  

These are a very important and valuable observations. We acknowledge that we did not include a control to rule out the possibility of mouse protein contamination from the 0.8% mouse OP9 cells, giving our previous experience in detecting low-abundance proteins.

Our experiments were conducted using a Q-TOF mass spectrometer operating in Data Dependent Acquisition mode, which biases the selection of peptides towards those with the strongest signal, compromising the detection of low-abundance peptides. Our experience analyzing human or plant cells infected with intracellular bacteria on this mass spectrometer shows that very few bacterial proteins are detected and typically detected with only one peptide, resulting in invalid identifications (valid identifications need at least 2 peptides). However, the identification of 67 mouse proteins with 2 or more peptides (as shown in Supplementary Table S1) suggests that these proteins were present within the human cells and not too low in abundance. We acknowledge that the absence of a control to rule out the possibility of mouse protein contamination from the 0.8% mouse OP9 cells is a weakness of the study and we address this issue in the discussion section (lines 315-320 of the version with tracked changes). Nevertheless, we believe it is unlikely that the identification of 67 mouse proteins, which represent 4.2% of the total number of valid identifications in the co-cultured CCRF-CEM cells, resulted solely from 0.8% contamination by mouse OP9 cells

Regarding the enrichment of human cells by FACS, we deeply appreciate your suggestion. In our experience, and based on some published results, the purity of sorted cells using FACS ranges from 96 to 99 % (for example, DOI: 10.3791/1546 or DOI: 10.3389/fimmu.2021.686530). Thus, we believe that this methodology, or others using magnetic beads, would not completely rule out the possibility of detecting mouse proteins as a result of the presence of a small amount of contaminant OP9 cells. Just as FACS, our harvest procedure turned out to be efficient. We made a new Figure 1 showing the results of three independent replicates of co-cultured CCRF-CEM cells, and the mean and standard deviation of Sca-1 negative cells (0.7 ± 0.5 %) are mentioned in line 314 of the version with tracked changes.

The suggestion to use fluorescently tagged proteins in mouse cells is highly valuable, as it would provide direct evidence of protein transfer from mouse to human cells. Although we plan to conduct this experiment, its implementation will require a significant amount of time. Nevertheless, we want to emphasize that our preliminary study using shotgun proteomics has yielded promising results, suggesting the intercellular transfer of proteins in our 3D co-culture system. We acknowledge that further validation using other techniques is necessary, but we believe that our methodology represents an initial step towards detecting intercellular protein transfer using a simple experiment. As such, we have revised the manuscript to avoid making definitive statements about protein exchange, while acknowledging the preliminary nature of our findings. Please refer to lines 308-309, 387 and 418-421 of the version with tracked changes.

Figure 2: The authors nicely showed that co-cultured cells are more resistant to insults. Indeed, this validates the 3D culture system as a novel and useful tool to study drug resistance in cancer.

Thanks for your comment.

Figure 4: The authors validated two proteins identified from the proteomics analysis, albeit with the caveat laid out above. While the authors nicely include three replicates in the figure, they are quite variable. I believe adopting the revised setup will alleviate this inconsistency.

Although 3D culture systems are attractive due to their ability to better reproduce cell-to-cell interactions, its use for methodologies that require a high number of cells, involves the need to process several batches of laborious handling. We believe that this contributed to the variable results. Despite implementing several steps to process the batches in the same way, the metabolic state of the 3D co-cultured CCRF-CEM was consistently heterogeneous, resulting in variability in Western blot experiments, enzymatic activity determinations (Figure 5), and the separation of corresponding dots in the Principal Component Analysis (new Supplementary Figure S1). This effect was not observed in monocultures. Nevertheless, the detection of both proteins, TSPO and galectin-3, in three independent biological replicates can be interpreted in the context of intercellular transfer or proteins as a yes/no experiment. From this perspective, the results in Figure 4A are in line with the data from our proteomics analysis. While we recognize the inherent variability in in vitro systems, we are confident that this variability did not detract from the main objective of our study.

Round 2

Reviewer 3 Report

In the original manuscript, the reviewer questioned that the murine proteins detected in the murine-human cell co-culture by proteomics were false positive signal from the experiments. In the revised draft, the authors illustrated and articulated sufficiently that this was not a concern. Therefore, the reviewer has no further suggestions and agrees to publish this manuscript.